# Optimization of a Field Emission Electron Source Based on Nano-Vacuum Channel Structures

**DOI:** 10.3390/mi13081274

**Published:** 2022-08-08

**Authors:** Ji Xu, Congyuan Lin, Yongjiao Shi, Yu Li, Xueliang Zhao, Xiaobing Zhang, Jian Zhang

**Affiliations:** 1School of Electronic and Information Engineering, Nanjing University of Information Science & Technology, Nanjing 210044, China; 2Joint International Research Laboratory of Information Display and Visualization, School of Electronic Science and Engineering, Southeast University, Nanjing 210096, China; 3Wenzhou Key Lab of Micro-nano Optoelectronic Devices, Wenzhou University, Wenzhou 325035, China

**Keywords:** nano-vacuum channel structure, electron source, field emission, simulation, electric properties

## Abstract

Recent discoveries in the field of nanoscale vacuum channel (NVC) structures have led to potential on-chip electron sources. However, limited research has reported on the structure or material parameters, and the superiority of a nanoscale vacuum channel in an electron source has not been adequately demonstrated. In this paper, we perform the structural optimization design of an NVC-based electron source. First, the structure parameters of a vertical NVC-based electron source are investigated. Moreover, the symmetrical NVC structure is further demonstrated to improve the emission current and effective electron efficiency. Finally, a symmetrical nano-vacuum channel structure is successfully fabricated based on simulations. The results show that the anode current exceeds 15 nA and that the effective electron efficiency exceeds 20%. Further miniaturizing the NVC structures in high integration can be utilized as an on-chip electron source, thereby, illustrating the potential in applications of electron microscopes, miniature X-ray sources and on-chip traveling wave tubes.

## 1. Introduction

Researchers have been utilizing nano-fabrication techniques to organize sub-100 nm vacuum nanogaps since the conception of nanoscale vacuum channel (NVC) was established [1], demonstrating that these devices can perform in applications of fast response and high frequency [2,3,4,5]. On the other hand, the reliability of NVC under high temperature and radiation conditions has conjointly been confirmed [6,7]. However, most studies in this field have only focused on optimizing the structure parameters or emitter materials [8,9,10,11], e.g., shortening the vacuum channel or utilizing low-dimensional materials, while few studies have reported on possible future applications, such as inverse functional circuits or complementary triodes [12,13,14]. The superiority of nanoscale vacuum channels in specific applications has not been adequately demonstrated.

Canon reported, back in 2006, an array-type surface conduction electron source (SCE) [15]. Each array cell has a carbon film with nanoscale vacuum gaps, and a voltage applied at both ends can obtain a horizontally conducted emission current. Furthermore, the electron source emission current can be obtained by applying a high voltage in the vertical direction. The core structure of SCE is essentially a carbon-film-based NVC structure, which has the advantages of low operating voltage, fast response time and compatibility with semiconductor processes [16,17,18]. Therefore, further miniaturizing the NVC structures in high integration can be utilized as on-chip electron source, illustrating its potential in applications of electron microscopes, miniature X-ray sources and on-chip traveling wave tubes.

In this paper, we perform structural optimization design of NVC electron source by simulations and experiments. First, the on-chip NVC-based electron source is designed. A planar NVC structure is used as the emission region, where the electrons are “pulled” out by the electric field force in vertical direction through an external anode. Moreover, the symmetrical NVC structure is further investigated to improve the emission current and effective electron efficiency. The symmetrical nano-vacuum channel structure is successfully fabricated based on simulations. Finally, the results show that the anode current exceeds 15 nA and that the effective electron efficiency exceeds 20% at a gate voltage of 6 V and an anode voltage of 300 V. The current density is correspondingly estimated.

## 2. Materials and Methods

### 2.1. Simulation

The software used for the simulation is CST Studio 2020, with the module of Particle in cell. The material used in the simulation is a perfect electric conductor (PEC), which is a material with infinite electrical conductivity σ, where there is only current on the surface and no magnetic flow and where the surface electric field is perpendicular to the surface at all times. The environment of the simulation is set in a complete vacuum, eliminating the influence of environmental factors on the simulation.

### 2.2. Fabrication of NVC

For the cathode and gate in the experiment, we chose gold as the material. Gold has a strong electrical conductivity and stable chemical properties with a work function of 5.1 eV. The fabrication processing includes spin coating of photoresist, electron beam exposure, thin film deposition and lift-off process of the gold electrode. First, the substrate was precleaned successively in acetone solution, anhydrous ethanol and deionized water for 30 min using an ultrasonic cleaner, while blow drying with a nitrogen gun. 

Secondly, the substrates were baked on a heating table at 90 °C for 90 s to remove the residual water vapor and adequately maintain the cleanliness of the oxide wafer surface. Then, the substrate was spin-coated with PMMA at 600 rpm for 5 s and at 3000 rpm for 30 s. After spin-coating, the substrate was placed on a heating table and baked at a temperature of 180 °C for 90 s to ensure the photoresist curing. The thickness parameters of the photoresist can be controlled by adjusting the parameters of the spin-coating.

In addition, the photoresist on the substrate was exposed by an electron beam, where the exposure dose was 600 μC/cm2 and the electron beam current size was 100 pA. After that, the substrate was developed in a 1:3 ratio of 4-methyl-2-pentanone to isopropanol for 120 s, and then transferred to isopropanol for fixing for 60 s. Moreover, the electron beam evaporation process was conducted with gold used as the emitting and collecting material for NVC, followed by a subsequent lift-off process. 

The final step was the post-treatment process of the device, where the sample was thermally annealed at 400 °C in a tube furnace under hydrogen (70 sccm) and argon (40 sccm). In this case, the hydrogen was used to remove the residual photoresist, and the post-annealing can also improve the strength of the gold electrode, making it more adhesive to the substrate.

### 2.3. Characterization

The FEI Quanta 200 Scanning Electron Microscope (SEM) was applied to characterize the microscopic morphology of the nanoscale vacuum channel structure. 

## 3. Results and Discussion

### 3.1. Simulation of an NVC-Based Electron Source

The reported surface conduction electron source applies a low voltage to both ends of the nano-vacuum channel to form a horizontal electric field [15,16,17,18,19]. Then, the cathode emits electrons, forming a transverse conduction current. When a voltage is applied to the anode in the vertical direction to form a vertical electric field, some of the electrons emitted from the cathode will be “pulled” by the electric field force to the anode, which forms the emission current of this electron source. Thus, the basic structure can be reduced to a nano-vacuum channel structure in the plane and a vertical anode to collect electrons. The advantage of such design is that the vacuum channel structure can achieve field emission at low voltages and a portion of the electrons emitted from the field can be “pulled” by the electric field perpendicular to the vacuum channel, forming the emission current.

The on-chip nano-vacuum channel-based electron source was designed based on the above design in simulation, as shown in Figure 1. We found that the cathode and the gate of the planar electron source were on the surface of the substrate, while the anode of the vertical electron source was overhanging directly above the substrate. The width of the cathode and the gate structures was 0.5 μm, and the thickness was 0.2 μm. The spacing between the cathode and the gate was 100 nm, forming the NVC structure. The cathode emits electrons to the gate forming a gate current when it applies a voltage to the gate. The anode is 1 μm away from the nano-vacuum channel structure, and the anode current can be formed by applying a voltage to the anode.

Figure 2 shows the variation curve of the output current with the gate voltage for different anode voltages for the on-chip NVC-based electron source. The anode voltage has a greater effect on the anode current, and it has a smaller effect on the gate current. Secondly, the anode current undergoes three stages of change as the gate voltage increases. In the first stage, the anode current gradually rises for the anode current growth region as the gate voltage increases. 

In the second stage, the gate voltage continues to rise while the anode current ceases to rise and remains stable, which is the anode current saturation region. Moreover, when the anode current transitions from the growth zone to the saturation zone, the gate voltage varies with different anode voltages. In the third stage, the anode current decreases sharply as the gate voltage increases further. This is the anode current cut-off zone, and the gate voltage varies with different anode voltages when the anode current begins to fall.

The electrons in the vacuum channel are mainly subjected to the electric field force in two directions: the transverse electric field (E_t_) between the cathode and the gate and the electric field (E_⊥_) generated by the anode perpendicular to each other (as shown in Figure 3). Thus, the three phases of the anode current change can be analyzed by the change of the electric field force on the electron transport. In the anode current growth region, the surface electric field strength of the cathode emitting surface and the field emission current are negligible due to the relatively small gate voltage. 

At this time, both the number of electrons generated by field emission and the number of electrons “pulled” to the anode by E_⊥_ are small, forming a weak anode current. The surface electric field intensity of the cathode emission surface rises and the number of electrons generated by field emission becomes larger as the gate voltage increases. The number of electrons “pulled” to the anode by E_⊥_ also becomes larger, and thus the anode current increases. In the current saturation region, while the gate voltage increases leading to a larger surface electric field intensity at the cathode emitting surface, the number of electrons generated by field emission also becomes larger. 

However, it is difficult for more electrons to be “pulled” to the anode by E_⊥_, due to the increase in lateral E_⊥_. The increase in the number of electrons at this stage creates a relative balance with the increasing E_t_, leading to the anode current remaining essentially constant. The number of electrons transported to the anode by E_⊥_ decreases sharply, and the anode current drops sharply. When the gate voltage is large enough, the electrons emitted from the cathode field are intercepted by the gate, and the anode current becomes zero as shown in Figure 3b. In addition, as the anode voltage increases, the current saturation zone moves in the positive direction of the two axes. In other words, a higher anode current is obtained at a higher gate voltage.

Table 1 shows the gate current, anode currents and effective electron efficiency of electron sources at different gate voltages at 50 V. The effective electron efficiency of a nano-vacuum channel electron source is the ratio of the current transported to the anode to the cathode current, which also corresponds to the ratio of the anode current to the sum of the anode and gate currents and can be expressed by the following equation.
(1)$$\eta =\frac{I_{A}}{I_{A}+I_{G}}$$
where η is the effective electron efficiency rate, $I_{A}$ is the anode current, and $I_{G}$ is the gate current. Table 1 shows that the anode current of the vertical type is about 0.69 μA at a gate voltage of 10 V and an anode voltage of 50 V.

Figure 3a shows that the electrons transported to the anode are emitted to the top of the vacuum channel. Thus, the number of electrons collected by the anode is related to the width of the cathode. The greater the width, the more electrons will be transported to the anode. According to Equation (1), the effective electron efficiency rate reflects the proportion of the anode current, which is essentially the proportion of the number of electrons transported to the anode. 

Thus, the effect of the gate current needs to be considered when designing the structure. E_t_ rises for the gate voltage increase at the same anode voltage, but E_⊥_ almost does not change. The effective electron efficiency rate declines, which makes the ratio of the number of emission electrons to the total number of emitted electrons decrease. This phenomenon indicates the direction of improving the effective electron efficiency of electron sources: designing nano-vacuum channel structures with low gate voltages and high anode currents. 

In addition, according to the electron trajectory diagram shown in Figure 3a, most of the electrons are still intercepted by the gate, forming the gate current. To reduce the gate current, we further optimized and adjusted the thickness of the cathode. Figure 4 shows the variation of the anode current and the effective electron efficiency rate for different cathode thicknesses of the electron source, at an anode voltage of 50 V and a gate voltage of 10 V. 

The anode current starts to increase with increasing cathode thickness and remains constant when the cathode thickness exceeds about 0.15 μm. The cross-sectional area of the cathode increases with increasing cathode thickness, resulting in an increase in the number of electrons emitted. However, the effective electron efficiency tends to decrease monotonically with increasing cathode thickness. The effective electron efficiency rate can exceed 50% at a cathode thickness of 0.01 μm, which is about five-times higher than the 9.87% shown in Table 1.

### 3.2. Design of a Symmetrical NVC-Based Electron Source

The size of the anode current is positively correlated with the cathode emission current. Therefore, the cathode emission current must be increased to obtain a higher anode current. According to field emission theory [20,21,22], there are several ways to increase the cathode emission current, such as shortening the vacuum channel length, reducing the cathode tip radius of curvature and increasing the emission area. For example, the operating voltage can be reduced by designing a cathode structure with small tip radius of curvature [23,24,25,26].

In addition, designing multiple cathode arrays is a straightforward and effective way to increase the emitting area [27,28,29,30,31]. Based on this idea, a symmetrical nano-vacuum channel electron source is designed as shown in Figure 5. The nano-vacuum channel structure has multiple cathode tips, each of which has a radius of curvature of 500 nm. The distance between each cathode tip and the gate is 100 nm, while the thickness of the gate and the cathode is 100 nm. 

Figure 6 shows the variation curves of the emitter current with the gate voltage for anode voltages of 5 V, 10 V and 15 V, respectively. It can be noted that three typical anode current variation stages are exhibited. Compared to the single flat cathode emitter, the anode current increases by two orders of magnitude (from 10^−8^ A magnitude to 10^−6^ A magnitude) by designing multiple curved cathode emitters.

Figure 7 shows the electron trajectories at different gate voltages for an anode voltage of 5 V. We note that all electrons would be intercepted by the gate and cannot be transported to the anode after the gate voltage increases to a certain value. In addition, the different electric performance of the electron source at different anode and gate voltages were calculated separately, as shown in Table 2. 

It can be seen that the anode current increases with the increasing gate voltage at the same anode voltage, which reduces the effective electron utilization of the source. Second, the increase in anode voltage improves the anode current at the same gate voltage and the effective electron utilization rate of the electron. Among them, the anode current can reach a maximum value of 12.58 μA at a gate voltage of 11 V and an anode voltage of 15 V. 

Furthermore, the effective electron efficiency can exceed 22%, which is comparable to some of the existing reported on-chip electron sources while requiring a higher operating voltage [32,33,34]. Additionally, the electron emission efficiency in tunneling electron sources (TESs) can reach 87.0%, which can be combined with the NVC structure in our future work [35]. The simulation work in this section shows that field emission electron sources based on nano-vacuum channel structures have great potential for on-chip applications.

### 3.3. Measurement of the Symmetrical NVC-Based Electron Source

Based on the simulation results in the previous section, we found that the symmetrical NVC structure had potential as a high-performance electron source. Considering the complexity of the preparation process for practical testing, we simplify the symmetrical nano-vacuum channel structure. A symmetrical NVC structure was prepared using electron beam lithography as shown in Figure 8. The gate radius is 1 μm, and the cathode tip is 100 nm from the gate, while the radius of curvature of the cathode tip is about 250 nm. The field emission electrical properties were tested in a vacuum SEM vacuum chamber using a nano-manipulator, as shown in Figure 8c. To start with, we tested the field emission from the multiple cathode array to the gate, which is on the same plain. 

Figure 9a,b shows the results of the field emission properties from multiple cathode array to the gate, and its corresponding electron source test at a gate voltage of 6 V with different anode voltages. It can be seen that the nano-vacuum channel structure starts to emit current at a gate voltage of about 1.5 V, and the field emission current exceeds 10 nA when the gate voltage is increased to 6 V. The inset shows the corresponding Fowler-Nordheim (F-N) fit curve, demonstrating that the surface conducting current follows the field emission process [36,37]. 

When conducting field emission electron source tests in the vertical direction, the gate voltage is kept at 6 V, and the anode current value is measured with a step of 0.2 s. The anode current rises with increasing anode voltage. When the anode voltage is 300 V, the gate current is approximately 10~15 nA, the anode current is more than 5 nA, and the effective electron efficiency is about 20%. The cathode emission current region is a rectangular region of 0.5 μm in length and 0.35 μm in width, as shown in Figure 10. To estimate the cathode emission current density, the emission area could be obtained by analyzing the F-N curve, and the emission area is related to the slope and intercept. The F-N equation can be further simplified as [36]:(2)$$J=\frac{A\beta^{2}E^{2}}{\phi }exp\left(-\frac{B\phi^{3/2}}{\beta E}\right)$$
where β is the field enhancement factor, A = 1.54 × 10^−6^ and B = 6.83 × 10^7^ are constants. Substituting J = I/s and E = *β* × V into Equation (2), Equation (2) can be deduced as:(3)$$I=aV^{2}exp\left(-\frac{b}{V}\right)$$
(4)$$a=\frac{sA\beta^{2}}{\phi }$$
(5)$$b=-\frac{B\phi^{3/2}}{\beta }$$
where s is the emission area (m^2^), ϕ is the work function of the material (eV). Take the logarithmic transformation of Equation (3), Equation (6) is the commonly used F-N fitting curve that determines whether the tunneling process occurs.
(6)$$ln\left(\frac{I}{V^{2}}\right)=lna-\frac{b}{V}$$
where a and b are the constants that can be obtained from the slope and intercept of F-N fitting curve, respectively. As a result, according to the inset of F-N fit curve in Figure 9a, we further substitute the calculating a and b into Equations (3) and (4), so that the effective emission area of the cathode tips can be estimated as 6.4 × 10^−9^ m^2^. The estimated values of current density in the region are shown in Table 3. 

Furthermore, the anode and gate current show the magnitude of 10 uA obtained by simulations, while the experiment results are barely at the magnitude of 10 nA. We assumed that the materials used in the simulation process and the external environment are ideal, that the performance of the actual environment (Au as the electrodes and low vacuum degree (10^−3^ Pa) in experiment) would be inferior to the simulation results. However, the experiment data are consistent with the trends obtained by simulation. 

In future work, we would optimize the structure parameters of the simulations to closely match the practical test. The gate is used as a control electrode to modify the anode current. When adjusting the gate and anode voltages, we also avoid the gate collecting large amounts of electrons. In our opinion, the perpendicular electron beam would be suitable for an electron source. With an optimized gate and anode voltage, the emitting electrons would be a perpendicular electron beam. In addition, the anode here is used for detecting the order of electron beam, and thus a focused electrode may replace the anode in an actual electron source.

The results show that the emission current rises and the effective electron efficiency of the electron source improves by designing a symmetrical nano-vacuum channel structure with multiple cathode tips. The emission current is still relatively low in values, while the current density is considerable. Future work can continue to increase the emission area or prepare an emission array structure in order to increase the emission current that applied in practical devices.

## 4. Conclusions

In this paper, an optimization of field emission electron source based on nano-vacuum Channel Structures was performed, and the performance parameters were simulated and tested. The test results showed that, when the gate voltage was 6 V and the anode voltage was 300 V, the anode current exceeded 5 nA. The effective electron utilization was over 40%, and the current density was estimated.

The electron generation mechanism of devices with a nano-vacuum channel structure as their core is field emission, which has the advantages of a fast response and high current density. In addition, due to the minimal electrode spacing, current can be emitted at low voltages, thereby, improving the stability of the cathode. Coupled with its own small size structure, this has the potential to miniaturize and integrate electron sources, leading to promising applications, such as small electron microscopes, miniature X-ray sources and on-chip traveling wave tubes. 

Furthermore, advances in modern information technology have placed greater demands on the response speed and high-frequency characteristics of devices, and nano-vacuum channel structures are an option to meet the direction of device development. Nano-vacuum channel structures can operate properly under extreme conditions, such as radiation and high temperatures, due to their inherent properties of a vacuum, which is an advantage that solid-state devices do not yet have.

## Figures and Tables

**Figure 1 micromachines-13-01274-f001:**
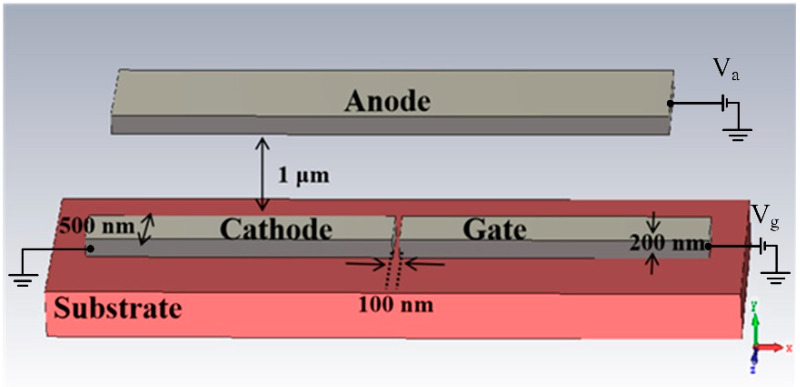
Schematic diagram of the NVC-based electron source.

**Figure 2 micromachines-13-01274-f002:**
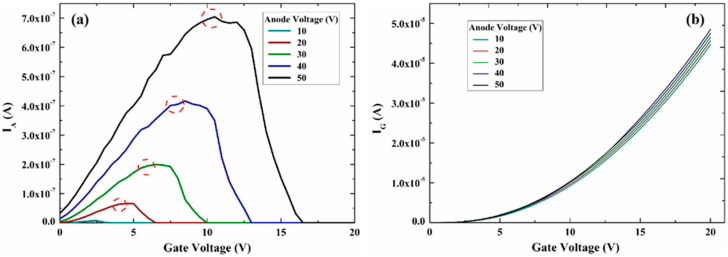
Simulation of the electric performance of the NVC-based electron source, (**a**) the anode current versus the gate voltage and (**b**) the gate current versus the gate voltage.

**Figure 3 micromachines-13-01274-f003:**
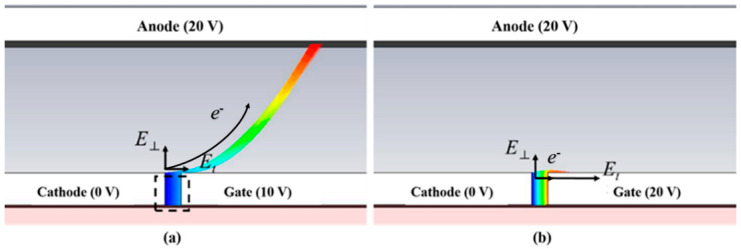
Top view of the electron trajectory of the NVC electron source at different gate voltages. (**a**) 10 V. (**b**) 20 V.

**Figure 4 micromachines-13-01274-f004:**
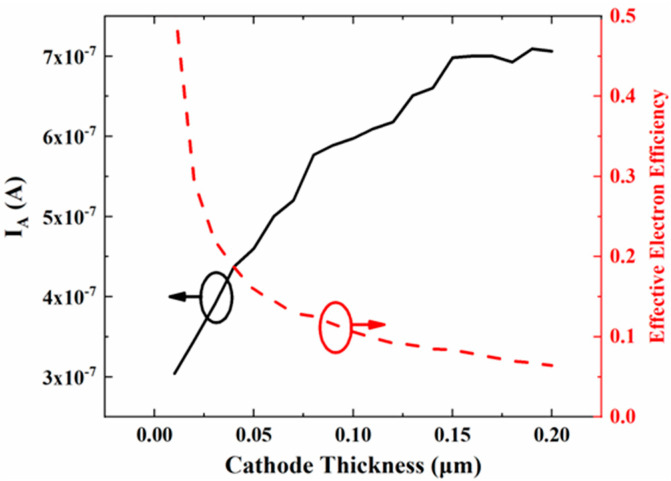
Optimization of the effective electron efficiency by decreasing cathode thickness.

**Figure 5 micromachines-13-01274-f005:**
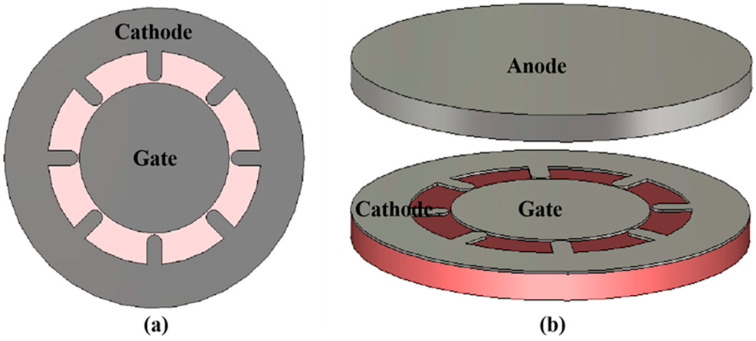
(**a**) Top view of the symmetrical structure. (**b**) Three-dimensional schematic.

**Figure 6 micromachines-13-01274-f006:**
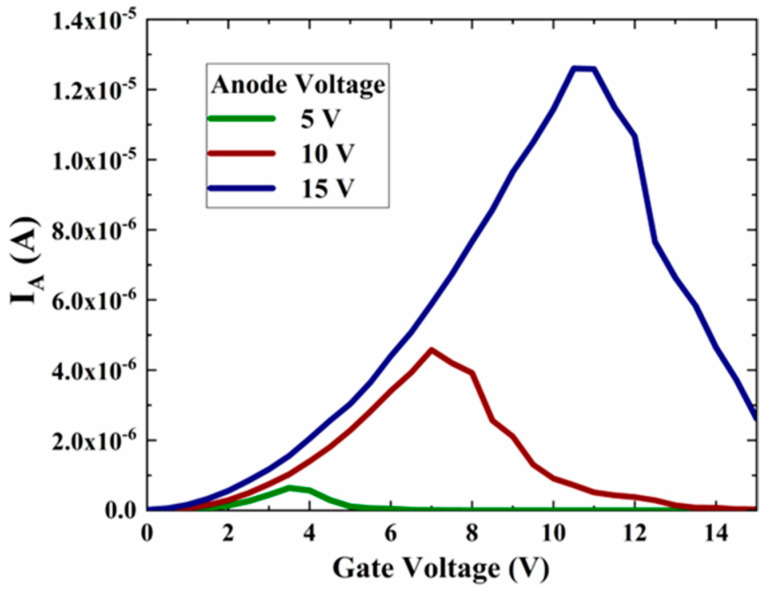
Variation curves of the emitter current with the gate voltage for different anode voltages.

**Figure 7 micromachines-13-01274-f007:**
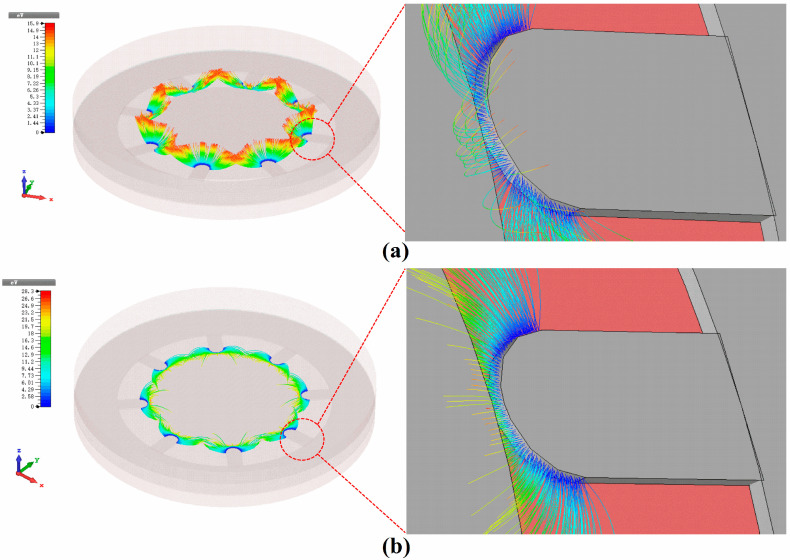
Electron trajectory at fixed anode voltage with different gate voltages: (**a**) 3 V and (**b**) 8 V.

**Figure 8 micromachines-13-01274-f008:**
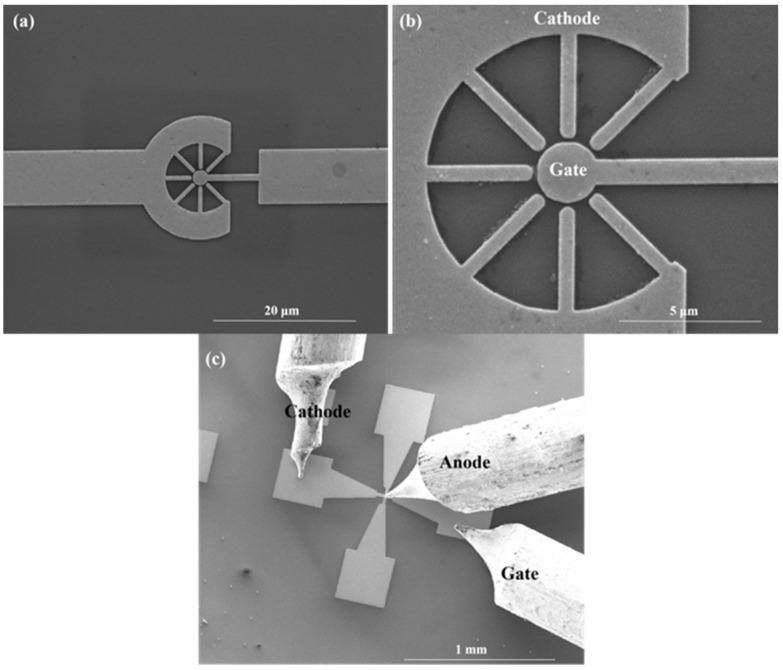
SEM images of (**a**) a top view of a symmetrical NVC-based electron source, (**b**) zoom-in of the nano-vacuum channel structure and (**c**) electrical measurement in a vacuum SEM vacuum chamber using a nano manipulator.

**Figure 9 micromachines-13-01274-f009:**
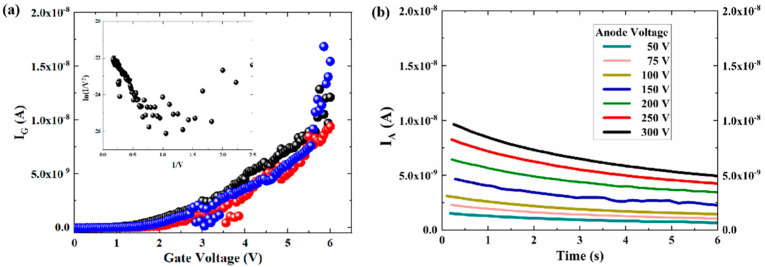
(**a**) Field emission characteristics, inset shows the corresponding F-N fit curve. (**b**) Emission current of electron source with different anode voltage at a fixed gate voltage of 6 V.

**Figure 10 micromachines-13-01274-f010:**
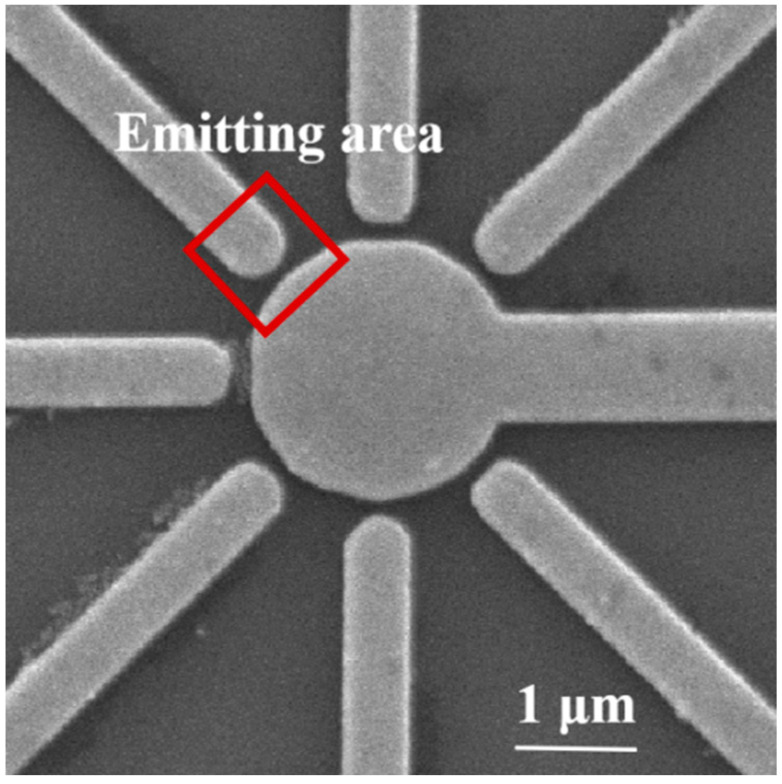
SEM image of the emitting area of the NVC structure.

**Table 1 micromachines-13-01274-t001:** Performance of the NVC electron source at a fixed anode voltage of 50 V.

Gate Voltage (V)	Gate Current (μA)	Anode Current (μA)	Effective Electron Efficiency
2.5	0.30	0.21	40.67%
5	1.94	0.40	17.12%
7.5	5.23	0.57	9.87%

**Table 2 micromachines-13-01274-t002:** Performance parameters of electron sources at different anode voltages and different gate voltages.

Anode Voltage (V)	Gate Voltage (V)	Gate Current (µA)	Anode Current (μA)	Effective Electron Efficiency
5	3	1.14	0.44	27.85%
	4	3.07	0.57	15.66%
	5	6.67	0.11	1.62%
10	3	1.31	0.75	36.41%
	4	3.09	1.40	31.18%
	7	14.03	4.58	24.61%
15	3	1.41	1.17	45.35%
	4	3.28	2.05	38.46%
	11	44.46	12.58	22.05%

**Table 3 micromachines-13-01274-t003:** Electric characteristics for the emitting area.

Gate Voltage (V)	Anode Voltage (V)	Cathode Current (nA)	Anode Current (nA)	Cathode Current Density (A/m^2^)
6	300	>15	>2.50	>2.34

## Data Availability

Not applicable.

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
