# Peer review of "Optimization of a Field Emission Electron Source Based on Nano-Vacuum Channel Structures"

_micromachines, 2022, doi:10.3390/mi13081274_

Round 1
Reviewer 1 Report
The paper presents the results of the nano-vacuum channel (NVC) emitter, from the design (including simulations) to the fabrication and testing of the NVC emitter. The structure is planar which is not a novel idea, and actually, the authors said it in their paper, however interesting are the performance results, especially in the triode configuration. Therefore, I recommend publishing but if the authors complete their manuscript with a few more stuff:
1. What about Paschen's Law? In micron and sub-micron gaps, in a vacuum, the break-down voltage can be very low.
2. I miss the data concerning the tools, and methods, e.g. including the simulation software or EL machine, etc., to be used to design and fabricate the NVC emitter. The same for materials to be used.
3. Why the simulated object is different in size, i.e. the gate, than the fabricated one? I am not sure if there are made correctly without the right method description. For example, if the FEM was used, it seems the size of the mesh is too large compared to the tip - generally, it would be fair to show the meshing and a type of element chosen for the mesh.
4. Similarly, as well as if the Gate voltage exceeds 11 V compared to 15 V of the anode, it might be that the gate emits electrons rather than the anode... the question is what materials the author used for the gate and cathode.
5. Fig. 9a the FN plot - the straight line is overinterpreted, and again, if the emitting material is a sort of nano-carbon material, the dots point right trend but the straight line is not correct then.
5. Table 2 description is missing!
6. Why there are various values for anode current between table 2 and figure 9b?
7. The authors in conclusions pointed: "The electron generation mechanism of devices with a nano-vacuum channel structure as their core is field emission, which has the advantages of fast response, no heating, and high current density.":
a) how did you calculate the current density? Please show the calculation.
b) why do you claim there is no heating on what bases? You do not show any results proofing that.
Besides, nicely, logically set the data and description.
Reviewer 3 Report
The article presents a structural optimization design of nanoscale vacuum channel electron source. The article is incomplete, the method section is missing, which make the manuscript difficult to understand. Below I have more comments about this issue and suggestion how to improve the manuscript.- Introduction
- Methods
- No method section?
- How the simulation were done?
- Results and discussion
- page 3 Figure 1, could you add the field direction and electrons trajectories?
- Could you give the definition of anode current and gate current.
- page 4, lines 130-131, Could explain (and add reference) why: the number of electrons collected by the anode is related to the width of the cathode
- Figure 4 is not discussed in the text
- Table 2 does not have a title
- Figure 8 no information in the fabrication process.
- Can you directly compare the simulation results with the experimental measurement?
- The simulations were done with an anode voltage less than 50 V, but the measurement were done at 300 V, why?
- How this design could be used as an electron source, if all electron are collected by the anode or the gate?
- Conclusions
- The conclusions is too general, it does not highlight the results in the manuscript.
Round 2
Reviewer 2 Report
The questions and comments have been responded point by point and the full text has been addressed. However, there still some crucial issues need to be corrected before accepted:
(1) Perfect electric conductor is still used for the simulation, which results in a unreliable large current as the author claims. As the author describe, golden is used for the measurement. For my opinion, the parameters of golden should be used for the simulation to see if there has a good match between simulation and measurement results.
(2) I am still strongly concern about the emission area. In field emission, the surface area can not be regarded as the emission area (Sg), especially when the current density is a crucial factor for valuing the performance of device. The emission area could be obtained by analyze the F-N curve and the emission area is related to the slope of intercept. The authors should understand the F-N theory and use it to analyze the data.
Reviewer 3 Report
- The methods section could still be improved
- Reference for the software CST, version and module used?
- A small discussion between simulation and measurement should be added like you have done in the response letter.
- Idem for how it could be used as electron source.
Round 3
Reviewer 2 Report
All the issues have been addressed. I am pleased to recommend it to be published on your journal.